# Hybrid Modulation Strategy to Eliminate Current Distortion for PV Grid-Tied H6 Inverter

**Tao Yang** [1],*, **Xiaoxiao Hao** [2], **Ruoxu He** [1], **Zhen Wei** [1], **Tao Huang** [1] **and Yuzhi Zhang** [3]

[1]  WANMA Group, Hangzhou 310000, China; heruoxu@wanmagroup.com (R.H.);
    weizhen@wanmagroup.com (Z.W.); huangtao@wanmagroup.com (T.H.)
[2]  State Grid Jibei Electric Power Company, Beijing 100053, China; hao.xiaoxiao@jibei.sgcc.com.cn
[3]  ABB Inc, Raleigh, NC 27606, USA; yuzhi.zhang@us.abb.com
*   Correspondence: yangtao1987323@163.com; Tel.: +86-133-4581-0382

**Abstract:** This paper proposes a new hybrid modulation mode (HMM) to eliminate the zero-crossing distortion of grid current and enable reactive power provision for a H6 configuration PV (photovoltaic) grid-tied inverter. The common mode voltage, leakage current, and efficiency for the proposed approach are also analyzed. In order to improve grid frequency tracking a novel frequency self-adaptive proportional-integral-resonant (FSAPIR) controller is implemented which reduces error for changes in grid frequency. The proposed approach provides the basis for accurately adjusting the active and reactive current without error to improve the grid support capability of the inverter. Theoretical analysis, simulation, and experiment verify the newly proposed modulation mode and controller.

**Keywords:** proportional-integral-resonant; self-adaptive; common-mode leakage current; hybrid modulation; zero-crossing distortion; reactive power

## 1. Introduction

Photovoltaic power generation has become one of the key research subjects in renewable integration research, attracting great interest from both industry and academia. The grid-tied inverter plays a vital role in the energy conversion and control of the PV (photovoltaic) system [1]. With the proliferation of grid-tied PV generation and other distributed generation, the contribution which such generation can make to grid stability has come into sharp focus. Consequently, many countries are raising the grid code compliance standards for the grid-tied inverters. In Germany, the latest standard VDE-AR-N4105 [2] directs that the photovoltaic grid-tied inverter should have a strong active power regulation and reactive power compensation capability that can help stabilize the grid voltage and frequency [2]. Also, the inverter needs to quickly output the corresponding reactive power to support low-voltage ride through (LVRT) [3], to aid grid recovery and avoid a chain reaction [4].

Efficiency and cost are key drivers in the development of grid tied inverters, which has motivated the move towards the use of transformerless inverters. Although eliminating the transformer generally improves cost and efficiency, it can create the problem of common-mode leakage current [5] which can reduce the security of the entire system and result in serious EMI (Electromagnetic Interference) issues [6]. For the common full bridge (H4), non-isolated topology, this common mode leakage current is not an issue if a bipolar modulation is used. However, the use of bipolar modulation requires a large inductor and is typically less efficient than unipolar modulation. Under unipolar modulation the H4 topology experiences serious leakage current problems [7,8]. The problem of leakage current suppression when using unipolar modulation has been addressed by several approaches, typically either by modifying the inverter topology, or by introducing extra filtering to deal with it. The

recent paper by Barater et al. [9] gives a good overview of different approaches. For example, the German company SMA have developed the H5 topology [10] a configuration which can suppress the leakage current, although in that work the operation of the topology under non-unity power factors was not presented. The ability of the inverter to operate under a range of power factors is an important consideration under most grid connection codes. In 2013, Ji et al. [11] proposed the single-phase non-isolated H6 inverter utilizing a hybrid modulation. Experimental results validated that the leakage current issues in this H6 inverter are eliminated and the efficiency is also maintained. However, from the results presented in that paper, it appears the issue of grid current zero crossing distortion—due to a loss of control of the current in the region of the zero crossing—was not considered. As has been indicated by Kotsopoulos [12], the zero-crossing distortion can cause a resonant response between the grid impedance and the inverter filter capacitor which can cause voltage distortion that could cause potential disturbance to nearby electrical equipment. Also, the authors did not address the H6 inverter's non-unity power factor operation. The problem of common mode leakage can also be dealt with by the introduction of extra filtering and Barater et al. [9] recently showed how the simpler H4 topology with unipolar modulation, combined with an active filter can eliminate the leakage current and provide for reactive power. However, their results still show an efficiency penalty associated with the use of the active filter.

In summary, there are still improvements possible in the pursuit of a non-isolated inverter which satisfies the requirements of high efficiency, reactive power provision (free of current distortion) and elimination of common mode leakage currents. To address these requirements this paper presents a hybrid modulation scheme for a H6 inverter, which allows for high efficiency, elimination of common mode leakage currents, and provision of reactive power with low current distortion. It can also be shown that the hybrid modulation scheme can be applied to other topologies such as the H5 (topology name) and HERIC (topology name) [13].

The paper is structured as follows: Section 2 reviews the operation of the H6 inverter topology with Modulation Mode I (MMI) as previously proposed in [11] and illustrates why this modulation mode cannot output reactive current without zero crossing distortion. Section 3 introduces the new hybrid modulation scheme which can achieve reactive current output under the full power factor range without distortion. The application of the hybrid modulation scheme to the H5 and HERIC converter topologies is also discussed. Simulations and experimental results to validate the approach are presented in Sections 4 and 5 respectively. Finally, Section 6 discusses the conclusions of the paper and scope for future work.

## 2. H6 Grid-Tied Inverter Operation

Previously a unipolar modulation technique for the H6 inverter, which has good performance as regards common mode leakage current, was reported in [11]. Here we refer to this as modulation mode I (MMI). Figure 1a displays the H6 grid-tied inverter topology [14] and the detailed operation of this modulation mode I is shown in Figure 1b. $E$ represents the DC voltage which is regulated by the pre-boost converter from the PV generators, $V_{ab}$ is the terminal voltage of the inverter, $V_{ac}$ is the grid voltage, $i_L$ is the grid-tied current, $L_1$, $L_2$ represents the grid-tied inductor $L = L_1 + L_2$, $D_1$ and $D_2$ are the freewheel diodes, $S_1$–$S_6$ are the switches, and $G_1$–$G_6$ are the corresponding gate driving signals for the switches. Figure 2a–d shows the current path in the converter for different conditions. Figure 2a,b shown the operation for positive voltage and current and Figure 2c,d for negative voltage and current.

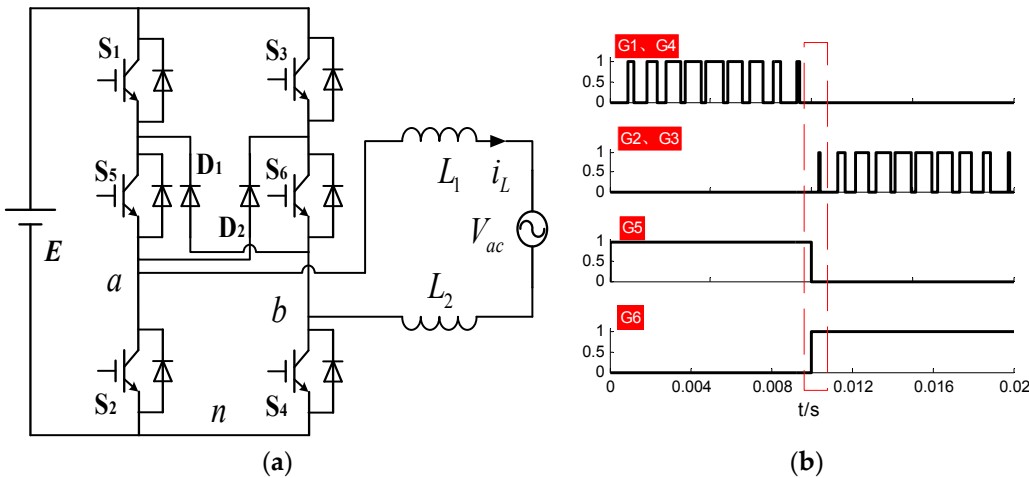

**Figure 1.** Structure of H6 and modulation mode I. (**a**) H6 grid-tied inverter topology, (**b**) modulation mode I (MMI).

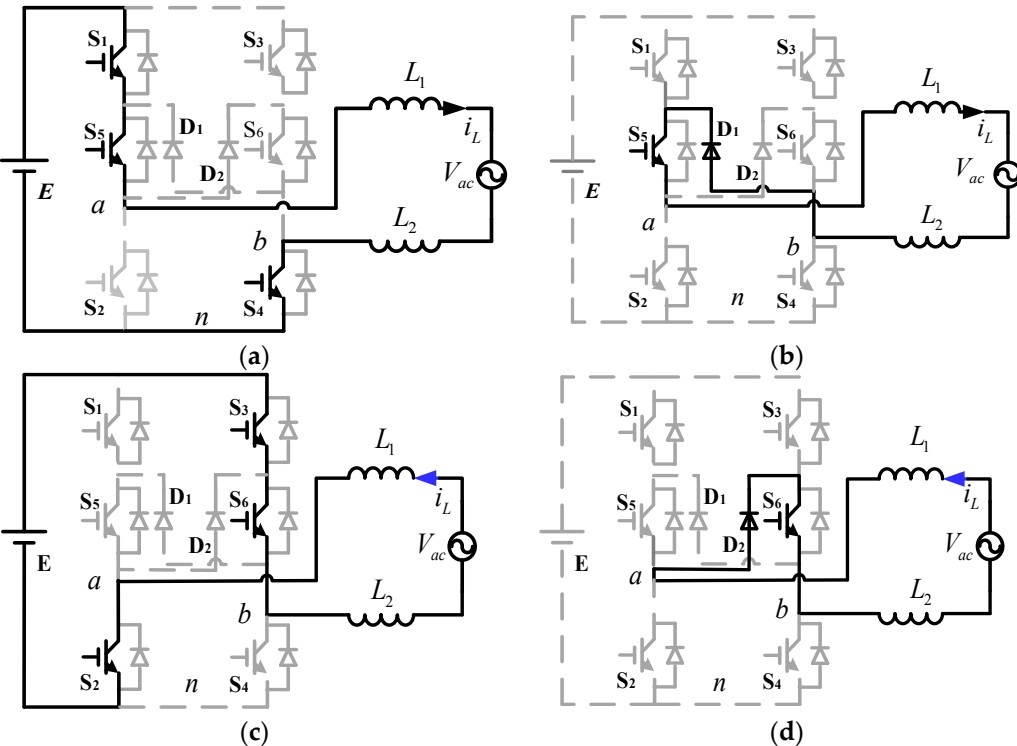

**Figure 2.** Switching states for modulation mode I. (**a**) Switch state 1, $i_L > 0$, S1 S4 S5 on, (**b**) switch state 2, $i_L > 0$, S1 S4 off, S5 on, (**c**) switch state 3, $i_L < 0$, S2 S3 S6 on, (**d**) switch state 4, $i_L < 0$, S2 S3 off, S6 on.

When the grid voltage is in the positive half cycle, S2, S3, and S6 are off, and S5 stays on. S1 and S4 switch on and off at high frequency with SPWM (sinusoidal pulse width modulation) control. When S1, S4 are turned on, see Figure 2a, the inductor is charging and the bridge output $V_{ab} = +E$. When S1 and S4 are turned off, see Figure 2b, the inductor current freewheels through D1 and S5 resulting in the bridge output voltage $V_{ab} = 0$. When the grid voltage is in the negative half cycle, S1, S4, and S5 are turned off, while S6 maintains remains on. S2 and S3 now switch on and off at high frequency using SPWM control. When S1 and S4 are turned on, see Figure 2c, the inductor is charging and the bridge output $V_{ab} = -E$. When S2 and S3 are turned off, see Figure 2d, the inductor current freewheels through diode D$_2$ and S$_6$ and the bridge output $V_{ab} = 0$.

*Grid Current Distortion and Reactive Power Output*

Although MMI can eliminate common mode leakage current, we show below that it has a problem with grid current distortion under reactive power output. The output voltage of a grid-tied inverter is clamped to the grid voltage and therefore can only regulate the active and reactive current by controlling the phase and amplitude of the grid-tied current. Thus, for reactive power output, there is necessarily a phase shift between inverter terminal voltage and current, the phasor relationship for which is illustrated in Figure 3.

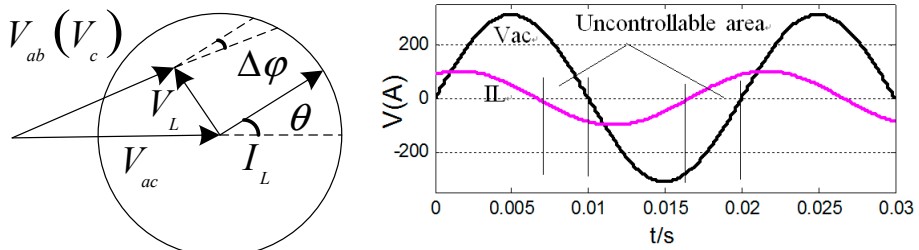

**Figure 3.** The diagram of vector and uncontrollable area.

Here $V_{ac}$ represents the grid voltage phasor, $V_L$ represents the inductor voltage phasor, $V_{ab}$ represents the bridge output voltage phasor, $I_L$ represents the current phasor (inductor current) of the grid-tied inverter. Based on the vector relationship, the phase angle difference between the inverter terminal voltage, $V_{ab}$ and its current $I_L$ for a non-unity power factor ($\cos(\theta)$), is given by

$$\Delta\varphi = \arctan\left[V_L \cos(\theta)/(V_{ac} - V_L \sin(\theta))\right] - \theta \tag{1}$$

Note that, even for the unity power factor at ($\theta = 0°$), $\Delta\varphi$ is not equal to zero.

For the H6 inverter under MMI, if the terminal voltage is positive and the current is negative and we continue to operate as in Figure 2a,b (turning on S5), the current cannot freewheel through D1 as normal when S1 and S4 are off, but rather flows through the parallel diodes of S1 and S4, so that the instantaneous terminal voltage is not 0 at this point, which from a control perspective means a loss of control, with resulting current distortion in this region.

The typical control model for the current control loop of a H6 grid-tied inverter is given in Figure 4, where, $i_L{}^*$ is the current reference, and $G_{r(s)}$ is the controller producing the control voltage, $V_c$. Note in order to offset the impact of grid voltage $V_{ac}$ for the inductor current control loop, the grid voltage feed-forward control is often adopted within the current loop. This grid voltage is added to the output terminal of the $G_{r(s)}$ controller in each switching cycle and this produces the control voltage $V_c$. This $G_{r(s)}$ controller is the traditional PI controller: $G_{r(s)} = K_P + K_I/s$, $K_P$ is the P control parameter and $K_I$ is the is the I control parameter.

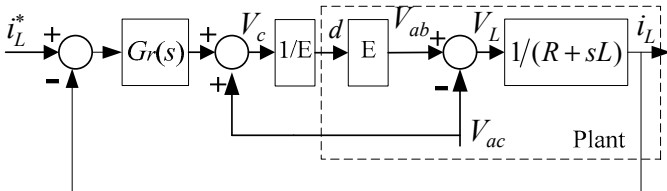

**Figure 4.** The control model of current loop.

Under the assumption of unipolar modulation, described as modulation mode I earlier, this control voltage is equal to the average terminal voltage $V_{ab} = dE$, where d is the duty cycle of switches S1 and S4 or S2 and S3. Inherent in this is the assumption that when S1 and S4 are off, (or S2 and S3 are off), the current freewheels through D1, S5 (or D2, S6), so that the terminal voltage transitions between

*E* and 0 (or −*E* and 0). However, when the current and voltage have opposite polarity, the inductor current cannot freewheel normally through either D1, S5 or D2, S6. Thus, the average terminal voltage $V_{ab}$ in this region of operation is not given by the assumption of unipolar modulation. The averaged value of $V_{ab}$ over one switching cycle therefore cannot equal the required reference voltage $V_c$, which means the inductor current is out of control and distortion of the current waveforms as illustrated in Figure 5 results. Note also that zero crossing distortion can exist in the grid current in the region shown in Figure 5 even for unity-power factor, because the current cannot reverse and is forced to operate in discontinuous conduction mode in the region of the zero crossing. However, the grid current distortion will be more pronounced when the grid-tied inverter outputs reactive current in non-unity power factor operation.

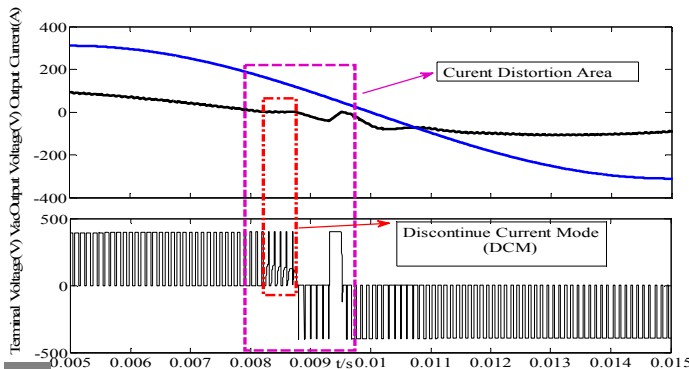

**Figure 5.** Terminal voltage during the distortion of current.

Thus, although the H6 inverter with MMI successfully eliminates common mode leakage current, it will have a problem with grid current distortion under conditions of reactive power provision. In the next section, we illustrate how this problem can be solved by introducing a new modulation approach in the region where voltage and current have opposite polarity.

## 3. Hybrid Modulation Mode under Non-Unity Power Factor

### 3.1. H6 Grid-Tied Inverter with Modulation Mode II

Modulation mode II (MMII), illustrated in Figure 6a is proposed to effectively suppress current distortion and achieve reactive power provision in the regions of operation where current and voltage are in opposite polarity. The operation of this modulation is illustrated in Figure 6. Take for example the case where the terminal voltage $V_{ab}$ is in the positive half cycle, but the current is in the negative cycle. In this situation, S1 and S4 are switched as before, however now S6 is switched in a complimentary manner to S1 and S4. This allow for a new freewheel path through D2, S6, when S1 and S4 are off, thus allowing the instantaneous terminal voltage to be zero and preserving the unipolar modulation. Conversely when the terminal voltage is in the negative half cycle, but the current is in the positive, switch S5 can be used with a complimentary switching to S3 and S2 to provide the necessary freewheel path.

This MMII could be used for the entire output voltage cycle, however doing so would incur the extra switching loss associated with the high frequency switching of S5 and S6. To avoid this MMI and MMII can be combined into a hybrid modulation mode (HMM) to solve the problem of current distortion while also minimizing the switching losses. This HMM consists of using MMI when the control voltage has the same polarity with the inductor current and using MMII in the region where the control voltage has the opposite polarity with the inductor current. This new modulation mode can be used for any range of power factor operation.

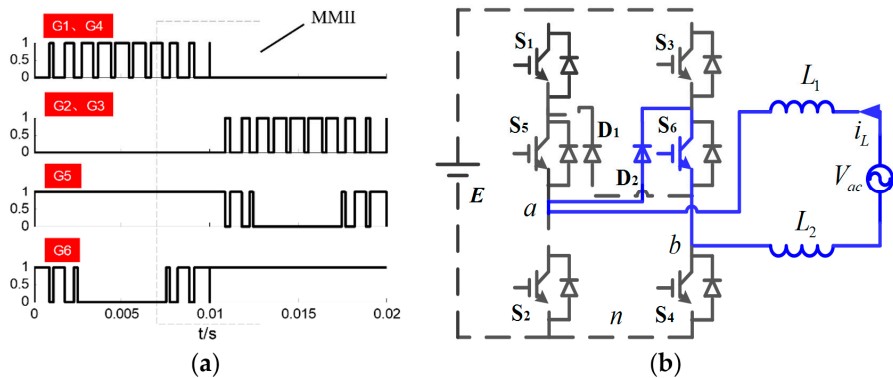

**Figure 6.** The novel hybrid modulation mode and operation mode 5. (**a**) The new Hybrid Modulation Mode (HMM), (**b**) operation mode 5.

For the H6 grid-tied inverter with HMM, the common-mode voltage can be analyzed for the six switching states. This analysis shows that the common mode voltage $V_{cm}$ can be maintained at a constant value 0.5 *E*, no matter which mode the system is in, and hence there is no common-mode leakage current present.

It is also worth noting that the H6 topology under MMI scheme does not need to insert a dead zone for the switches in the same phase leg. This is because the switches in the same phase leg are never all turned on during the same SPWM cycle with MMI [15]. In contrast however, under MMII, the DC voltage *E* will possibly be short circuited if an appropriate dead time is not inserted between the switches S1, S4, and S6. Although the instantaneous terminal voltage $V_{ab}$ cannot be freewheel to zero during the dead time interval, the dead time is very short compared to the whole switch period, therefore, there is almost no impact to the output AC voltage.

### 3.2. Hybrid Modulation Mode with H5 and HERIC

Apart from the H6 topology a number of other topologies have been developed which also allow for maintaining the common-mode voltage constant. These include the H5 topology from SMA Corporation [10] and the HERIC topology developed by the Sunways Company [16]. Similar to the H6-type inverter, both topologies can maintain a constant common-mode voltage with no common-mode leakage current for a single general modulation mode. The HMM technique presented here can also be extended to the H5 and HERIC topologies. As shown in Figure 7, utilizing the HMM, a loop for the freewheeling current can be formed for the non-unity power factor operation of both the H5 and HERIC converters thus eliminating the current zero-crossing distortion. The proposed HMM not only suits these three topologies, but also can be adopted in other single-phase topologies which need the freewheeling path and is required to eliminate the current zero-crossing distortion and provide reactive current.

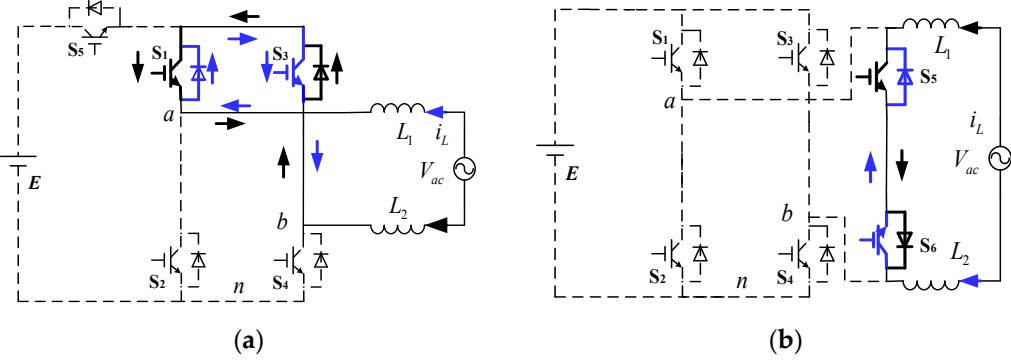

**Figure 7.** Freewheeling mode in the positive half period with the new hybrid modulation mode (HMM) for both unitary and non-unity power factor. (**a**) H5 grid-tied inverter. (**b**) HEIRC grid-tied inverter.

## 4. System Analysis and Control

### 4.1. Common-Mode Voltage and Leakage Current Analysis

Wang et al. [11] describes the common-mode voltage $v_{cm}$ of single-phase grid-tied inverter as

$$v_{cm} = 0.5(v_{an} + v_{bn}) - 0.5v_{ac} \tag{2}$$

and the leakage current as

$$i_{cm} = C\frac{dv_{cm}}{dt} \tag{3}$$

where, $v_{an}$ and $v_{bn}$ represent the pulse voltages between the two bridges midpoint and the DC bus negative terminal, respectively, which are varying at the inverter switching frequency. C represents the stray capacitance of PV module to ground. According to Equation (3), we can see the $i_{cm}$ is dependent on the $\Delta v_{cm}$. Note the component of common mode leakage current due to the grid voltage $v_{ac}$ can be ignored due to its low frequency of 50/60 Hz. Thus, the common-mode voltage can be approximately written as

$$v_{cm} \approx 0.5(v_{an} + v_{bn}) \tag{4}$$

The main common-mode leakage current $i_{cm}$ can be eliminated as long as the common-mode voltage $V_{cm}$ maintains a constant value. For the H6 grid-tied inverter with HMM, the common-mode voltage can be analyzed for five operation modes. This analysis shows that the common mode voltage $V_{cm}$ for this case can be maintained at a constant value $0.5 \times E$, no matter which mode the system is in, and hence there is no common-mode leakage current present.

### 4.2. Filter Inductor Calculation

The filter inductor is an essential component of the grid-tied inverter, with significant implications for the size and efficiency of the inverter, therefore we briefly present the selection of the inductance for the H6 topology under the proposed HMM. Assuming the switching frequency is $f_s$, the conduction duty cycle of switches is $d$, according to reference [17], it is clear to conclude that

$$\Delta i_L = \frac{[-(d-0.5)^2 + 0.25]E}{Lf_s} \tag{5}$$

when $d$ = 0.5, the ripple current $\Delta i_L$ will reach the maximum

$$\Delta i_{L\max} = \frac{E}{4Lf_s} \tag{6}$$

and this must be less than the allowable the current ripple $\Delta i_{LN}$.

As shown in [10], in order to connect to the grid with grid voltage $V_{ac}$, the DC voltage $E$ needs to meet the following conditions

$$E \geq \sqrt{2}\sqrt{V_{ac}^2 + (\omega L I_{LN})^2} \tag{7}$$

where, $I_{LN}$ is the RMS rated current. Combining Equations (6) and (7), we can attain the range of inductance for the H6 grid-tied inverter with the new HMM as

$$\frac{E}{4\Delta i_{LN}f_s} \leq L \leq \frac{\sqrt{E^2 - (\sqrt{2}V_{ac})^2}}{\sqrt{2}\omega I_{LN}} \tag{8}$$

As indicated in Reference [11], the inductance range for the H4 grid-tied inverter with bipolar modulation is

$$\frac{E}{2\Delta i_{LN} f_s} \leq L \leq \frac{\sqrt{E^2 - (\sqrt{2}V_{ac})^2}}{\sqrt{2}\omega I_{LN}} \tag{9}$$

Therefore, comparing Equation (8) with Equation (9), we can clearly observe that under the same current ripple conditions the H6 topology using HMM only requires half of the H4 inductance value, effectively reducing the overall size and weight of the system.

### 4.3. Adaptive Current Control under the Full Power Factor

The fluctuation of both voltage frequency and amplitude in a power system is inevitable. The common rule of "active power regulates frequency, reactive power regulates voltage" is often used in power systems to stabilize the frequency and voltage. Therefore, modern grid-tied standards always require that the grid-tied inverter should have the capability of zero error tracking of both reactive and active current set points. As indicated in [18], proportional integral (PI) controllers introduce a steady-state error in the tracking of a sinusoidal reference signal, although the proportional resonant (PR) controller can theoretically achieve zero steady-state error. However, the PR controller's performance can be very sensitive to any mismatch between its resonant frequency and the always slightly varying grid frequency. This can lead to a serious deviation from the expected control effect and can possibly lead to system instability. In order to improve the H6 grid-tied inverter performance in the regulation of frequency and voltage, we integrate the PI and PR controller in this paper and construct a new frequency self-adaptive proportional integral resonant (FSAPIR) controller. Figure 8 displays the self-adaptive current control loop, where the broken arrow indicates the frequency adaptive adjustment. It must be noted that a control delay of one switch period $T_s$ has been considered in this control loop.

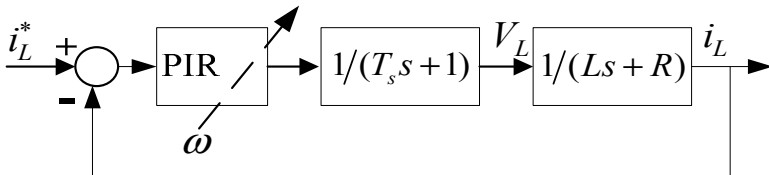

**Figure 8.** The self-adaptive current loop.

The transfer function $G_r(s)$ of FSAPIR regulator operates according to

$$Gr(s) = K_P + \frac{K_I}{s} + \frac{K_R s}{s^2 + \omega^2} \tag{10}$$

where, $K_P$ is the proportional gain, $K_I$ is the integration coefficient, $K_R$ is the resonant coefficient, and $\omega$ is the fundamental frequency of the grid. In order to realize the FSAPIR, see Figure 9, the improved Nikai Hiroyoshi integral frequency lock loop is used to obtain the grid frequency $\omega$ [19], and then its adaptive control adjusts the resonant frequency of the circuit.

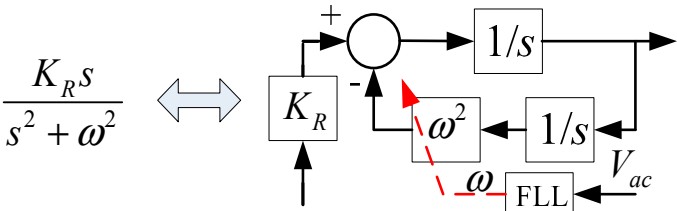

**Figure 9.** Self-adaptive frequency control.

Firstly, the resonance section of the circuit is set to zero, and then the tuned the PI controller parameters are obtained from $G_r(s) = K_P + K_I/s$. The open-loop transfer function is obtained from

$$G_O(s) = \frac{K_I(\frac{K_P}{K_I}s + 1)}{Rs(T_S s + 1)(\frac{L}{R}s + 1)} \tag{11}$$

Using the zero offset and ignoring the poles which are not conducive to the stability of the system results in $K_P/K_I = L/R$. From this we obtain the close loop transfer function of the PI controller with the expression

$$G_c(s) = \frac{\frac{K_P}{LT_s}}{s^2 + \frac{1}{T_s}s + \frac{K_P}{LT_s}} \tag{12}$$

The damping ratio is set as $\xi = \frac{\sqrt{LT_s}}{2T_s\sqrt{K_{ip}}} = 0.707$, the PI parameters are obtained from: $K_P = \frac{L}{2T_s}$, $K_I = \frac{R}{2T_s}$. The block diagram for the entire control scheme for the H6 inverter using the HMM and the FSAPIR approach is illustrated in Figure 10.

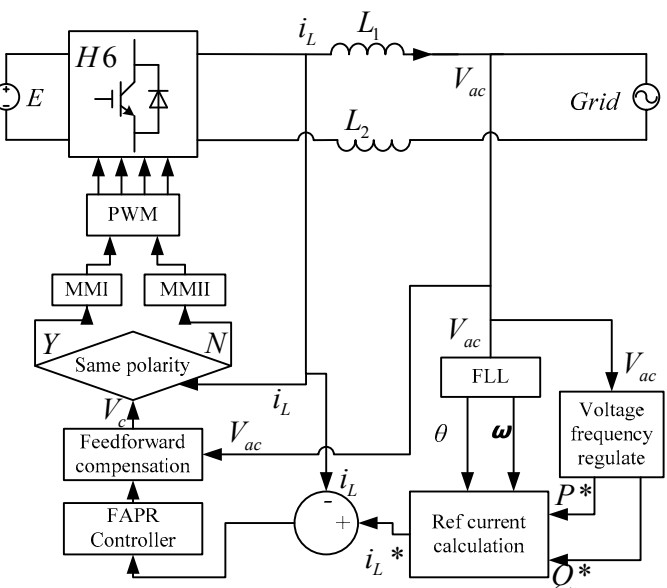

**Figure 10.** The block diagram of control with the new HMM.

## 5. Simulation Study

The simulation model is developed on the MATLAB/Simulink platform, using the Simulink S-Function module for realizing the control algorithm. The corresponding simulation parameters for the model are shown in Table 1.

**Table 1.** Parameters for simulation.

| Variables | Value | Variables | Value |
|---|---|---|---|
| Rated Power | 4.67 kVA | $T_s$ | 52 μs |
| Capacitor | 1000 μF | $f_s$ | 19.2 kHz |
| $E$ | 400 V | $L$ | 10 mH |
| $V_{ac}$ | 220 V | $f$ | 50 Hz |
| $\Delta i_{LN}$ | 15% Peak Current | $I_{LN}$ | 21.22 A |
| $K_P$ | 19.23 | $K_I$ | 1923.1 |
| $K_R$ | 19.23 | R | 0.2 Ω |

Figure 11 displays the inductor current, grid voltage and common-mode voltage of H6 inverter for two kinds of modulation modes in the scenario when the inductor current leads the grid voltage by 30°. As shown in Figure 11a, there is an uncontrollable region for modulation mode I and the current is seriously distorted in this region. It can be seen that the time duration of current distortion is larger than the time for which voltage and current have opposite polarity of $\Delta\varphi$, which is due to the system changing-over between a non-controlled region and a controlled area for which the current controller needs time to adjust. The results using the presented HMM techniques, are shown in Figure 11b. Clearly, adopting the new HMM for the H6 inverter system results in a controllable current with a more stable waveform characteristic.

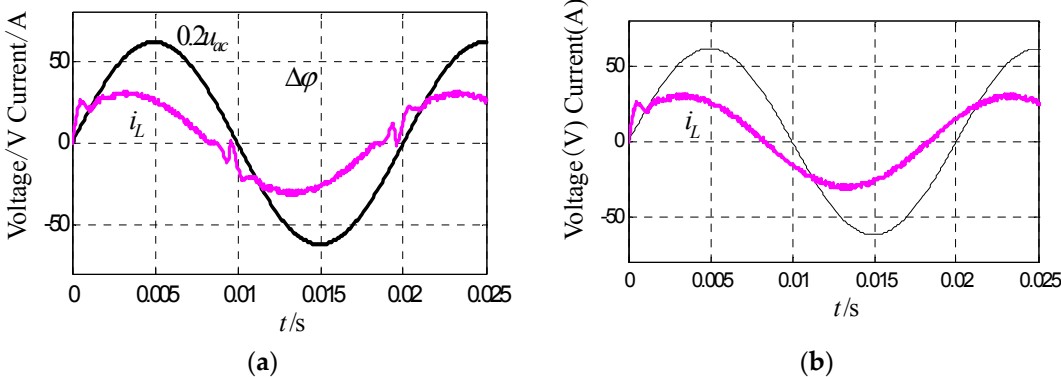

(a)　　　　　　　　　　　　　　　　　　　　　　　　(b)

**Figure 11.** Grid voltage and inductor current of H6. $\theta = 30°$, 5 times attenuation of voltage. (**a**) Modulation I. (**b**) Hybrid modulation mode (HMM).

Figures 12 and 13 show a similar comparison between the use of MMI and the new HMM, applied to the H5 and HERIC converter topologies. Again, it can be seen the new hybrid modulation mode eliminates the current distortion.

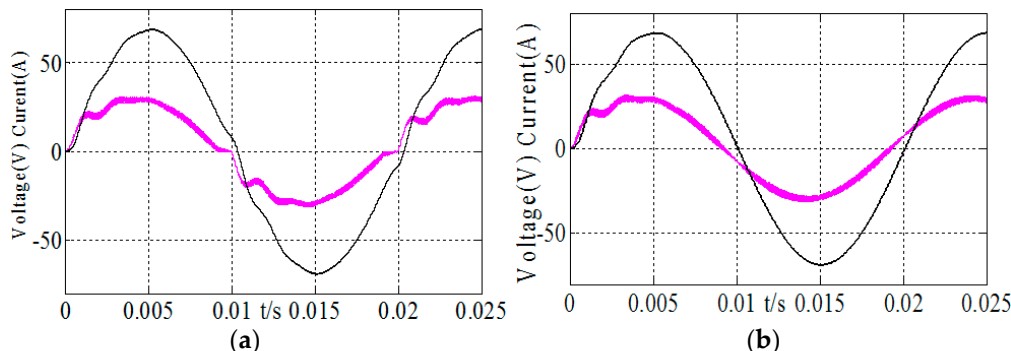

(a)　　　　　　　　　　　　　　　　　　　　　　　　(b)

**Figure 12.** The grid voltage and inductor current of H5. $\theta = 30°$, 5 times attenuation of voltage. (**a**) Modulation I. (**b**) Hybrid modulation mode (HMM).

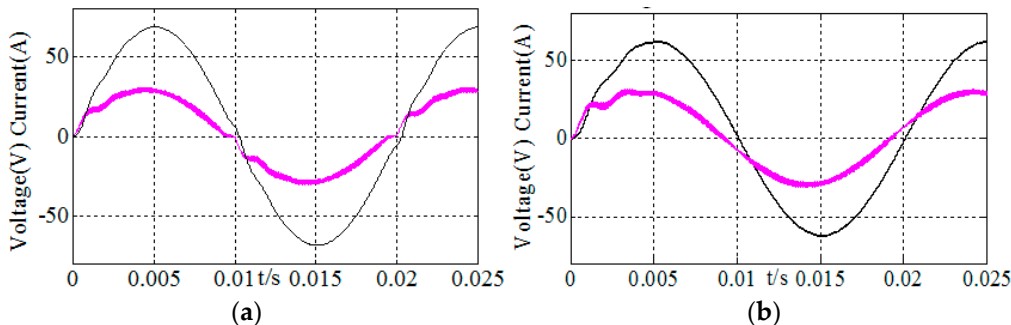

**Figure 13.** Grid voltage and inductor current of HEIRC. $\theta = 30°$, 5 times attenuation of the voltage. (**a**) Modulation I. (**b**) Hybrid modulation mode (HMM).

Figure 14 plots the common-mode voltage for the H6 converter using both modulation techniques. Note that for MMI as shown in Figure 14a the common-mode voltage is not a constant value in the region of opposite current and voltage polarity. However, when using the HMM the common-mode voltage becomes a constant value with no common-mode leakage current as displayed in Figure 14b.

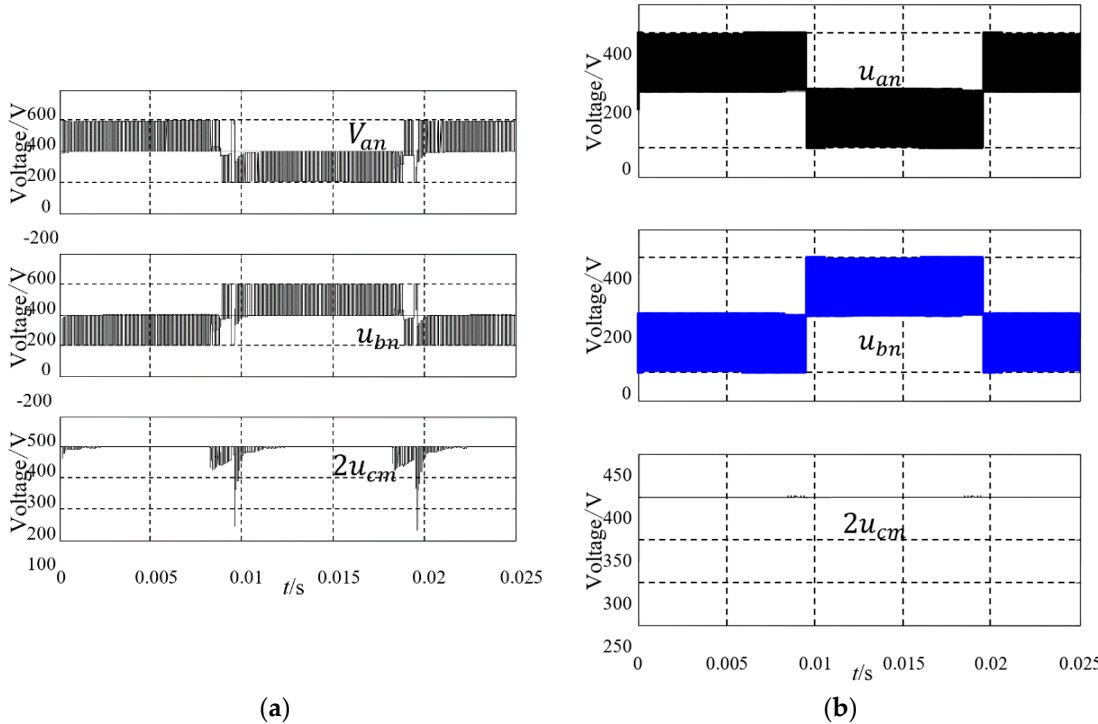

**Figure 14.** Common-mode voltage. (**a**) Common-mode voltage of modulation mode I, $\theta = 30°$. (**b**) Common-mode voltage of HMM, $\theta = 30°$.

Based on the system parameters from Table 1 earlier, the open loop and closed loop transfer function of the system under the PI, PR, and PIR controllers can be easily obtained. Both the open and closed loop bode plots for three controllers are shown in Figures 15 and 16 respectively. It should be noted that the closed loop bode plots in Figure 15 are obtained from the systems which have already been controlled. As can be seen from Figure 14, the gain at the resonant frequency under the PR controller is 300 dB. The drawback of PR control is that the gain at the low-frequency stage is lower than the PI control. However, zero steady-state error cannot be achieved while adopting PI control alone due to the gain of PI control at the grid frequency (the resonant frequency of the PR controller) being limited. At the high frequency stage, both the gain of PI and PR controllers drops at a rate of 40 dB per decade reducing the influence of the external noise on the system.

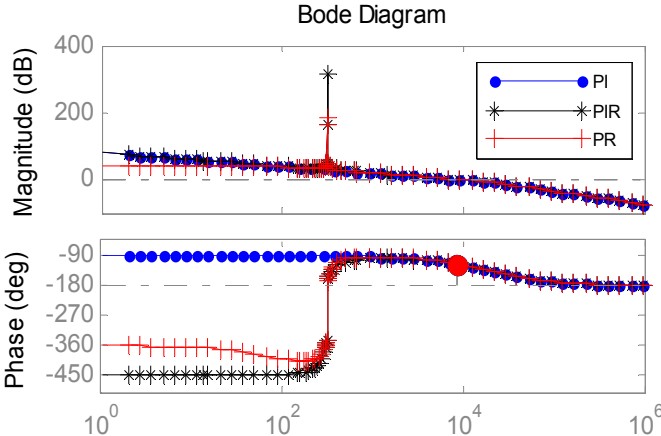

**Figure 15.** Bode diagram for open-loop. Frequency (rad/s).

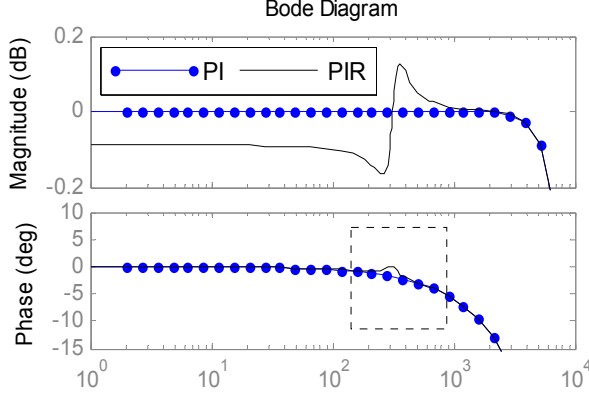

**Figure 16.** Bode diagram for closed-loop. Frequency (rad/s).

The PIR control method has been proposed to overcome the disadvantages of individual PR and PI control, with the system maintaining 65° phase margin and infinite gain margin—ensuring the system is sufficiently stable with the PIR controller [20]. Reviewing Figure 16 and Table 2, it is clear that there is a significant phase error under the PI control, resulting in a tracking error, but the proposed PIR control compensates for this deficiency.

**Table 2.** Comparison of errors.

| Controller/(50 Hz) | Magnitude/(dB) | Phase/(°) |
|---|---|---|
| PI | 0.00573 | −1.9 |
| PR | 0.0052 | −0.0146 |
| PIR | $-1.42 \times 10^{-6}$ | −0.0031 |

Figure 17 shows the output current $i_L$ and the tracking error $i_{err}$ when the frequency and phase of the reference grid voltage $V_{ac}$ changes from 50 Hz to 55 Hz, with a phase of 180° respectively at a period of t = 0.3 s. As we can see in Figure 17a, there is a large steady error after the frequency discontinuity event of the system under the non-adaptive PIR control; while for the system which adopts the FSAPIR controller, as shown in Figure 17b, the error can be eliminated in 0.05 s after the frequency discontinuity of the system.

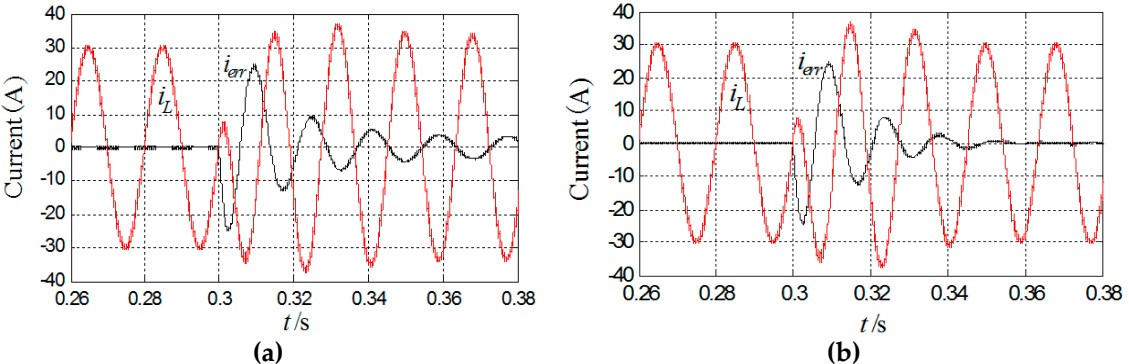

**Figure 17.** Inductor current and tracking error. (**a**) Non-adaptive PIR control. (**b**) FSAPIR control.

## 6. Experimental Validation

In order to further verify the preceding analysis and test the validity of the proposed design, a 4.67 kW prototype has been designed and built. The DC voltage $E$ is provided and regulated at 400 V by a pre-boost converter [1]. A picture of the laboratory prototype is depicted in Figure 18, the main parameters and devices of the inverter prototype are provided in Table 3.

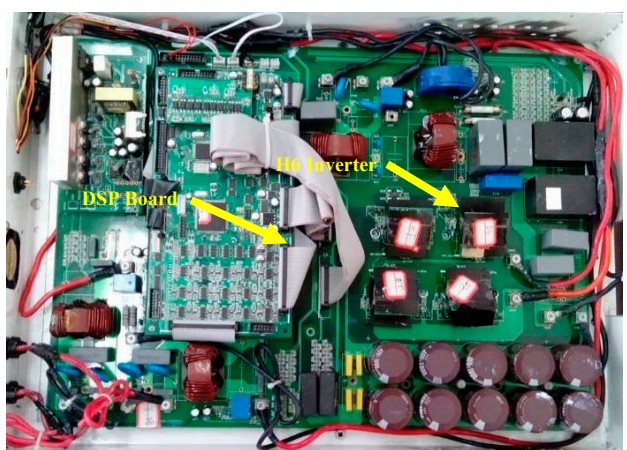

**Figure 18.** The prototype of H6 grid-tied inverter.

**Table 3.** Parameters of the experimental prototype.

| Parameter | Value | Parameter | Value |
|---|---|---|---|
| Rated power | 4.67 kW | Capacitor | 1000 μF |
| DC bus voltage | 400 V | Inductors | 2 mH |
| Switching frequency | 19.2 kHz | S1 to S4 | CoolMosIPW60R045CP |
| Grid voltage | 220 V/50 Hz | S5 to S6 | IKW75N60T(IGBT) |
| D1 to D2 | 60EPU06PBF | DSP chip | TMS320F2802 |

Figure 19 shows the terminal voltage of switches and inductor current of the H6 inverter for MMI when the power factor angle $\theta = 0°$. It can be seen that current distortion does indeed exist in the area of zero crossing. Figure 20 shows the grid voltage and inductor current of the H6 inverter utilizing the new HMM strategy for a power factor angle = 0°. As shown in Figure 20a, the current distortion is eliminated. Figure 20b shows that we adopt the MMII in a region of 10.8° around the zero crossing which is indicated by the drive pulses to S6 in this region. Figure 21 shows the voltage and current for non-unity power factor, $\theta = -30°$, for the H6 inverter with HMM. It clearly shows that there is no uncontrollable area therefore the issue of the current's distortion is effectively removed. Figure 22, verifies that the common-mode voltage is constant with very small fluctuations with leakage current of

21 mA. This is below the lowest level of leakage current of 30 mA where disconnection was necessary, as stated in the German DIN VDE 0126 standard [21].

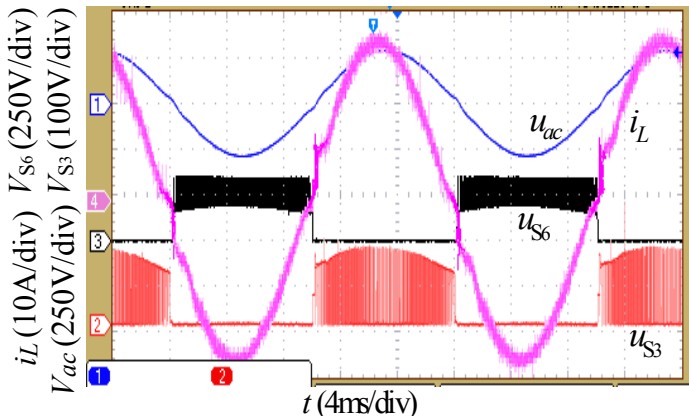

**Figure 19.** Terminal voltage of switches and inductor current MMI.

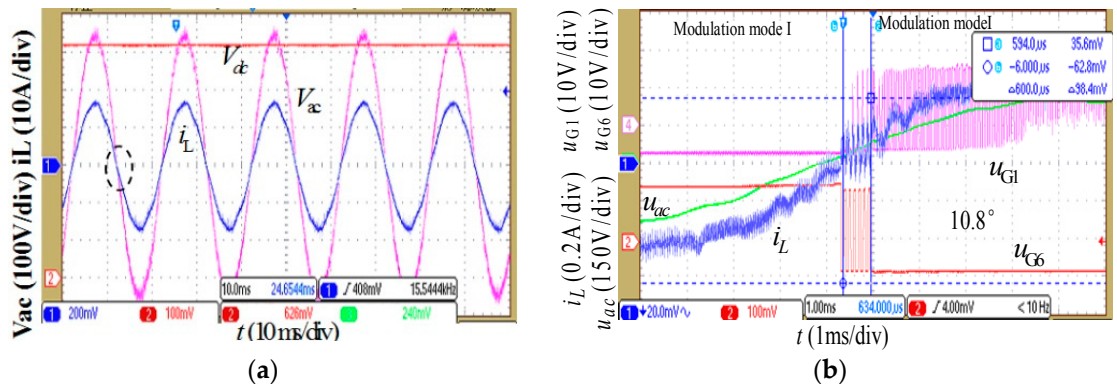

**Figure 20.** Voltage and inductor current under the HMM. (**a**) HMM: Voltage and inductor current, $\theta = 0°$. (**b**) HMM: Zoom in during the zero-crossing area, $\theta = 0°$.

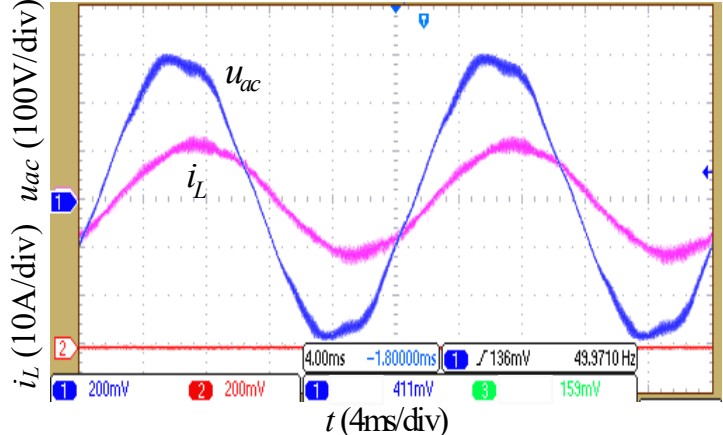

**Figure 21.** Voltage and inductor current under the HMM for 30° phase shift between voltage and current. HMM: Grid voltage and current, $\theta = -30°$.

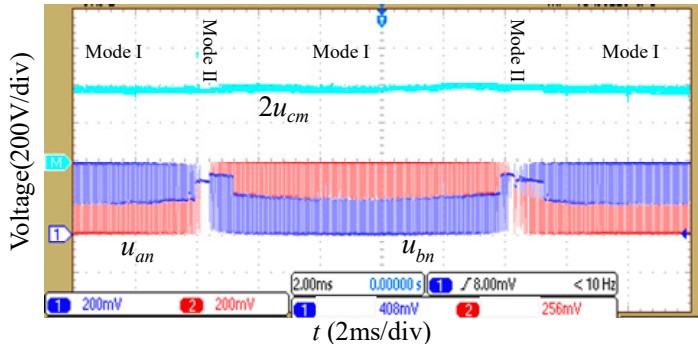

**Figure 22.** Common mode voltage. HMM: 2 times the common-mode voltage, $\theta = -30°$.

In order to validate the performance of the proposed FSAPIR controller, the grid voltage with frequency variation has been simulated with a Chroma grid simulator. Figure 23 shows the experimental waveforms of the system under FSAPIR control when the frequency of current reference signal is 55.11 Hz with amplitude of 30 A. It can be seen from Figure 23 that the real current can track the reference signal with low error and the total harmonic distortion (THD) is 2.53%, which is substantially less than 5% requirement of the IEEE929-2000 standard [22].

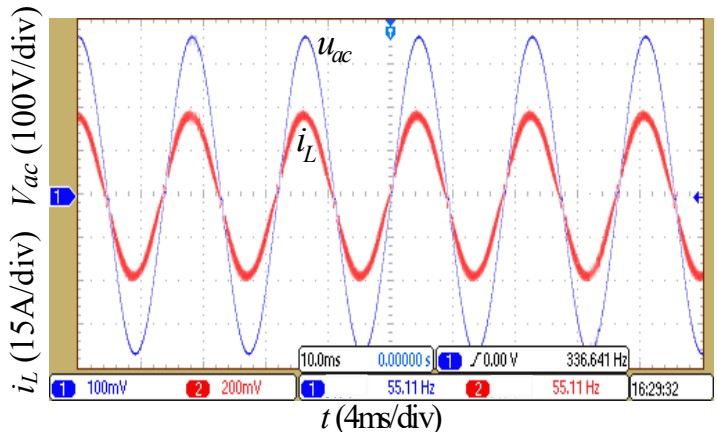

**Figure 23.** Voltage and current under the FSAPIR controller.

In the H6 implementation with HMM a separate diode is used to replace the body-diode of the switches with a positive impact on efficiency. Figure 24 shows the power efficiency curve; it can be seen that the efficiency of the system can achieve a maximum 98.2% when the system is operating at 50% of the power rating. According to further calculation, we can calculate the European efficiency as defined in [22] as 97.9%. For comparison Figure 24 also shows the efficiencies for the H4, H5, and HERIC topologies. These converter efficiencies are obtained from measurements. The H6, H5, and HERIC configuration efficiencies are higher than H4 [11,23,24]. Therefore, the HMM H6 inverter can be an inverter with a high efficiency and the capability of reactive power outputting.

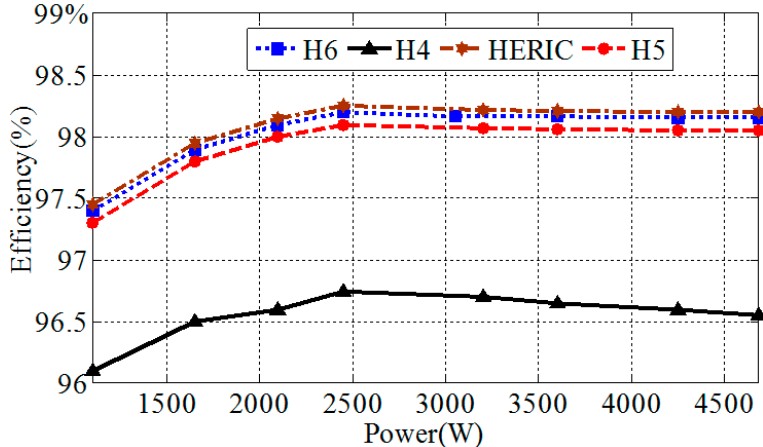

**Figure 24.** The efficiency of inverters.

## 7. Conclusions

This paper has presented a novel hybrid modulation technique to enable the H6 grid-tied inverter to operate with low distortion under non-unity power factor. The simulations and measurements validate the approach and demonstrate that high efficiency is also possible. The core of the proposed HMM is that the bridge can still output unipolar SPWM voltage waveform for the H6 inverter topology with no common-mode leakage current, while still maintaining current control and low distortion under all output power factors. It has been shown that when the MMII has been adopted in the uncontrollable region, it has improved the waveform profile of the grid-tied current and has reduced the THD. The modulation techniques can also be applied to other common inverter topologies. Based on the assumption that frequency fluctuation are inevitable on the grid with high penetrations of distributed energy resources, this paper proposes a new frequency self-adaptive PIR (FSAPIR) controller for the H6 grid-tied inverter and validated by experiments that the H6 grid-tied current can track the reference signal without error under this FSAPIR control. Thus, the inverter system can accurately regulate the active and reactive current.

**Author Contributions:** T.Y. and X.H. conceived the main idea and performed the tests, conducted data analysis, and wrote the manuscript. R.H., Z.W., and T.H. contributed to the test and simulation modelling. Y.Z. gave suggestions and revises.

**Funding:** This research was funded by the provincial key points Enterprise Research Institute of WANMA Group electric vehicle intelligent charging.

**Conflicts of Interest:** The authors declare no conflict of interest.

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
