# Peer review of "Hybrid Modulation Strategy to Eliminate Current Distortion for PV Grid-Tied H6 Inverter"

_applsci, doi:10.3390/app8122578_

Reviewer 1 Report

Summary:

In the submitted paper, the authors have presented a new Hybrid Modulation Mode (HMM) to eliminate the zero-crossing distortion of grid current with application to a PV inverter.

Recommendation: Minor revisions

Review: Overall the proposed approach is supported by the theoretical analysis, some simulated scenarios and finally experiments verification, which clearly shows the applicability and its potential.

- Its not clear if the proposed approach can be applied only to H6 inverter topology or it can be, even with modifications, applied to other type of inverters as well, besides the H5 and HERIC. Provide some comments about it.

- Provide some details about the controller Gr of Fig 4.

-Analyze how the hybrid modulation has affected in terms of time response the inverter operation.

-Provide the values of the tuning parameters (K) of the proportional resonant controller and PI controller.

-Overall a simulation comparison with the type of controllers that are usually used at PV inverters would support the potential of the proposed approach where 2 controllers are sequentially used.

-At Figs 12 and 13 there is a slight distortion before 0.005. Justify why the HMM has no affect over it.

Author Response

Point 1: Its not clear if the proposed approach can be applied only to H6 inverter topology or it can be, even with modifications, applied to other type of inverters as well, besides the H5 and HERIC. Provide some comments about it.

 Response 1: Actually, it was already talked in the section 1, paragraph 2 and paragraph 3, for the common full bridge (H4), non-isolated topology, this common mode leakage current is not an issue if a bipolar modulation is used, however the use of bipolar modulation requires a large inductor and is typically less efficient than unipolar modulation. Therefore, in this paper, we talk about the unipolar modulation, however under unipolar modulation the H4 topology experiences serious leakage current problems. The problem of leakage current suppression when using unipolar modulation has been addressed by modifying the inverter topology, such as H6, H5 and HERIC, therefore, the proposed approach can be applied only to H6, H5 and HERIC inverter.

 Point 2: - Provide some details about the controller Gr of Fig 4.

 Response 2: see line 129-131

 Point 3: - Analyze how the hybrid modulation has affected in terms of time response the inverter operation.

 Response 3: Actually, the time response of the inverter operation which is depend on the DSP calculation speed.

 Point 4: -Provide the values of the tuning parameters (K) of the proportional resonant controller and PI controller.

 Response 4: See Table.1 yellow part.

 Point 5-Overall a simulation comparison with the type of controllers that are usually used at PV inverters would support the potential of the proposed approach where 2 controllers are sequentially used.

 Response 5: See TABLE 2 and figure 17.

 Point 6-At Figs 12 and 13 there is a slight distortion before 0.005. Justify why the HMM has no affect over it.

 Response 6: Actually, slight distortion before 0.005 which is because the start-up situation. Therefore, the HMM has no affect over it.

Reviewer 2 Report

Manuscript presents a novel hybrid modulation technique for a H6 inverter, which allows operating with low distortion under non-unity power factor.

The work is found very interesting, relevant to the journal scope, well written and organized. In the background information is adequate to understand the aims and objectives of the study. The novelty of the study is made clear. The research design is appropriate. The simulations results together with the measurements validation are well presented and of great interest to the research community.

Author Response

Thanks for your time and suggestions.